# Casein Glycomacropeptide: An Alternative Protein Substitute in Tyrosinemia Type I

**DOI:** 10.3390/nu13093224

**Published:** 2021-09-16

**Authors:** Anne Daly, Sharon Evans, Alex Pinto, Catherine Ashmore, Anita MacDonald

**Affiliations:** Birmingham Women’s and Children’s Hospital, Birmingham B4 6NH, UK; evanss21@me.com (S.E.); alex.pinto@nhs.net (A.P.); catherine.ashmore@nhs.net (C.A.); anita.macdonald@nhs.net (A.M.)

**Keywords:** tyrosinemia type 1, protein substitute, amino acids, glycomacropeptide

## Abstract

Tyrosinemia type I (HTI) is treated with nitisinone, a tyrosine (Tyr) and phenylalanine (Phe)-restricted diet, and supplemented with a Tyr/Phe-free protein substitute (PS). Casein glycomacropeptide (CGMP), a bioactive peptide, is an alternative protein source to traditional amino acids (L-AA). CGMP contains residual Tyr and Phe and requires supplementation with tryptophan, histidine, methionine, leucine, cysteine and arginine. Aims: a 2-part study assessed: (1) the tolerance and acceptability of a low Tyr/Phe CGMP-based PS over 28 days, and (2) its long-term impact on metabolic control and growth over 12 months. Methods: 11 children with HTI were recruited and given a low Tyr/Phe CGMP to supply all or part of their PS intake. At enrolment, weeks 1 and 4, caregivers completed a questionnaire on gastrointestinal symptoms, acceptability and ease of PS use. In study part 1, blood Tyr and Phe were assessed weekly; in part 2, weekly to fortnightly. In parts 1 and 2, weight and height were assessed at the study start and end. Results: Nine of eleven children (82%), median age 15 years (range 8.6–17.7), took low Tyr/Phe CGMP PS over 28 days; it was continued for 12 months in *n* = 5 children. It was well accepted by 67% (*n* = 6/9), tolerated by 100% (*n* = 9/9) and improved gastrointestinal symptoms in 2 children. The median daily dose of protein equivalent from protein substitute was 60 g/day (range 45–60 g) with a median of 20 g/day (range 15 to 30 g) from natural protein. In part 2 (*n* = 5), a trend for improved blood Tyr was observed: 12 months pre-study, median Tyr was 490 μmol/L (range 200–600) and Phe 50 μmol/L (range 30–100); in the 12 months taking low Tyr/Phe CGMP PS, median Tyr was 430 μmol/L (range 270–940) and Phe 40 μmol/L (range 20–70). Normal height, weight and BMI z scores were maintained over 12 months. Conclusions: In HTI children, CGMP was well tolerated, with no deterioration in metabolic control or growth when studied over 12 months. The efficacy of CGMP in HTI needs further investigation to evaluate the longer-term impact on blood Phe concentrations and its potential influence on gut microflora

## 1. Introduction

Hereditary Tyrosinemia Type I (HTI), a rare inherited metabolic disorder of tyrosine metabolism, causes abnormal production of toxic metabolites (fumarylacetoacetate, maleylacetoacetate, succinylacetoacetate and succinylacetone), responsible for liver cirrhosis, renal tubular disease and vitamin D resistant rickets [1,2]. Treatment with nitisinone 2-(2-nitro-4-trifluoromethylbenzoyl)-1,3-cyclohexanodione (NTBC) prevents the accumulation of these toxic compounds by blocking the pathway at step two in the tyrosine pathway, mimicking HT III deficiency. This causes further elevation of blood tyrosine concentrations. To reduce blood tyrosine to an accepted treatment range of 200 to 400 µmol/L [3], a lifelong phenylalanine/tyrosine-restricted diet supplemented with a phenylalanine/tyrosine-free protein substitute is recommended. Additional phenylalanine supplementation is commonly necessary [4,5].

Traditionally protein substitutes are based on tyrosine/phenylalanine-free amino acids (L-AA), but due to the low prevalence of this condition, choices are limited; they are commonly powders or liquids only, with a narrow range of flavor choices. Patient acceptance is poor [6] and failure to take the prescribed dose adversely affects metabolic control and nutritional status [7].

An alternative protein substitute derived from modified casein glycomacropeptide (CGMP), low in tyrosine and phenylalanine but supplemented with additional amino acids, has been developed for HTI. The bioactive properties of CGMP are linked to the sialylated side chains attached to the macropeptide [8], and these are associated with clinical benefits, including improved gut flora [9,10], enhanced immune response [11] and anti-cariogenic properties [12]. A similar modified CGMP has been extensively studied in phenylketonuria (PKU). It is described as less bitter tasting and is linked to improved adherence in children [13,14,15]. There is also evidence that CGMP may improve growth and body composition in children with PKU [16]. In contrast, it is established that the residual phenylalanine content can lead to a deterioration in metabolic control in children with PKU [17]. There are no publications studying the outcome of patients with HTI when given CGMP as a protein substitute.

This two-part prospective interventional study aimed to investigate the acceptability, growth and metabolic control in a group of children with HTI given low Tyr/Phe CGMP (study product) as part of their protein intake over 12 months.

## 2. Materials and Methods

### 2.1. Study Design

This was a 2-part prospective intervention study (Figure 1).

Part 1: in a 28-day short-term acceptability and tolerance study, children replaced one or more of their usual daily doses of L-AA with the same amount of protein equivalent from the study product. Questionnaires on gastrointestinal tolerance, palatability and ease of use of L-AA and study product were completed by parents at baseline, week 1 and week 4 (see Appendix A for questionnaires). Weekly fasting blood samples were taken for tyrosine and phenylalanine. Target blood tyrosine ranges for children were 200 to 400 μmol/L as recommended by an international working group on HTI [3]. Weight and height were measured at baseline and day 28. 

Part 2: an extended follow-up study over 12 months was conducted in a cohort of children with HTI (*n* = 5) who chose to continue to use the study product (low Tyr/Phe CGMP PS). Regular (weekly to fortnightly) blood tyrosine and phenylalanine concentrations were measured. Weight and height were assessed at 6 and 12 months, and body mass index (BMI) was calculated. An acceptability evaluation was completed at the end of 12 months.

### 2.2. Subjects

Inclusion criteria: children diagnosed with HTI, ≥5 years of age and on a tyrosine/phenylalanine restricted diet, supplemented with L-AA and NTBC therapy.

Exclusion criteria: children <5 years of age, with a known allergy to milk, fish or soya (ingredients added to CGMP PS), additional co-morbidities (e.g., diabetes), non-adherent with dietary restriction or L-AA or participating in another study.

### 2.3. Protein Substitutes Used in the Study

The study product (*Tyr sphere*, a test product from Vitaflo International Ltd., Liverpool, UK) was a berry-flavored powder containing 11 mg of tyrosine and 36 mg of phenylalanine for each 20 g protein equivalent. It was reconstituted by adding 120 mL of water. It contained vitamins, minerals, trace elements and docosahexaenoic acid.

The L-AA supplements were either powders made with water (to a semi-solid consistency) or ready-to-drink liquids, both of which provided 10, 15 or 20 g of protein equivalent per dose. Products contained vitamins, minerals and trace elements, but docosahexaenoic acid was added to the Tyr Cooler only. Table 1 shows the nutritional profile of protein substitutes used.

### 2.4. Anthropometry

Height was measured using a Harpenden stadiometer (Holtain Ltd., Cyrmych, UK) and weight on calibrated digital scales (Seca, Medical Measuring Systems and Scales, Birmingham, Model 875, UK). Weight was measured to the nearest 0.1 g and height to the nearest 0.1 cm. All measurements were collected by one dietitian.

### 2.5. Blood Tyrosine, Phenylalanine, Succinylacetone and NTBC Concentrations

For each subject, the median tyrosine and phenylalanine blood concentrations were reviewed for 12 months prior to starting the study and compared with blood tyrosine and phenylalanine results collected during study parts 1 and 2. 

Information on routine biochemistry, plasma succinylacetone and NTBC concentrations were collected from routine clinic appointments.

Trained parents and caregivers collected early morning, weekly or fortnightly fasting blood samples on filter cards, Perkin Elmer 226 (UK standard NBS). All cards had a standard thickness, and blood tyrosine and phenylalanine concentrations were calculated on a 3.2 mm punch by (MS/MS) tandem mass spectrometry (Waters Xevo TQD, manufactured Wilmslow, Cheshire, UK). They were sent via first class post to Birmingham Children’s Hospital laboratory. 

### 2.6. Natural Protein Intake

All the children were prescribed a tyrosine/phenylalanine-restricted diet and the estimated median prescribed natural protein intake was 20 g/day (range 15 to 30 g).

### 2.7. Statistical Analysis

Descriptive statistics were used to analyze the data. Nonparametric paired Wilcoxon t-tests were used to compare anthropometry (weight, height and BMI), blood tyrosine and phenylalanine concentrations between the subjects. A *p*-value < 0.05 was considered statistically significant (analyzed using Prism GraphPad v 9.0).

### 2.8. Ethical Approval

The study was approved by the Northwest Liverpool East Research Ethics committee and granted a favorable ethical opinion, reference number 19/NW/0390 and IRAS (Integrated Research Application System) MCT-TYRS-2019-03-04. Written informed consent was obtained for all subjects from at least one caregiver with parental responsibility, and written consent was obtained from subjects if appropriate for their age and level of understanding.

## 3. Results

### 3.1. Subjects

In part 1, 11 children with HTI were recruited, and nine (4 females, 5 males) completed part 1. The median age was 15 years, range (8.6 to 17.7 years). Sixty-seven percent (*n* = 6) completed the full 28 days, three children completed up to 25 days and two withdrew on days 1 and 17.

Part 2, five children participated in the study part 2. The median age was 15.3 years (range 8.6 to15.4 years). There were three girls and two boys (Table 2).

### 3.2. Subject Withdrawal

One child (number 5) only completed study day one and did not progress any further, as the study product was disliked. Another child (number 6) developed viral gastritis and was unable to complete the study after day 17. A further four children chose to stop the study product at 28 days, preferring their original L-AA supplements.

### 3.3. Protein Substitute Type

Prior to starting the study, all children were taking L-AA supplements with added vitamins and minerals. Eight children were taking Tyr Cooler 20 (Vitaflo Int, Liverpool, UK), and three were taking powdered preparations: Tyr Express 20, (Vitaflo Int, Liverpool, UK), (*n* = 1), Tyr gel (Vitaflo Int. Liverpool, UK) (*n* = 1) and Tyr Shake and Go (Galen Ltd.) (*n* = 1).

#### Median Protein Substitute Intake in Part 1 and 2

In part 1 (*n* = 9), the median daily dose of protein equivalent from protein substitute was 60 g/day (range 45 to 60 g), providing 1.2 g protein equivalent/kg body weight (range 0.8 to 2.4 g/kg). The median amount of prescribed natural protein was 20 g/day (range 15 to 30 g), and no child was taking phenylalanine supplementation. In this short-term study, six children completely replaced their entire prescribed L-AA with the study product, while the remaining children changed one dose (20 g/day protein equivalent) of L-AA for the study product and took two doses of L-AA (40 g/day protein equivalent).

In part 2 (*n* = 5), the intake of protein equivalent from the study product was 60 g/day, with all children completely replacing their L-AA supplement with low Tyr/Phe CGMP PS. The median amount of protein equivalent was 1.0 g/kg (range 0.8 to 2.4 g/kg). The median intake of natural protein was estimated at 20 g/day (range 19 to 30 g).

### 3.4. NTBC and Concurrent Medications

All children took NTBC once daily. In parts 1 (*n* = 9) and 2 (*n* = 5), the median intake was 0.8 mg/kg (range 0.5 to 1.2 mg/kg), and no other medications were prescribed. Plasma succinylacetone concentrations were all <0.5 μmol/L (reference <0.5 μmol/L) [18].

### 3.5. Tyrosine and Phenylalanine Blood Concentrations

There were no statistically significant changes in blood tyrosine or phenylalanine concentrations over study part 1 (*n* = 9) or 2 (*n* = 5). In part 1, median blood tyrosine for the 12 months pre-study were 393 μmol/L (range 200–600 μmol/L) and post 28 days, 385 μmol/L (range 200–600 μmol/L) (*n* = 9). For the children who extended the study for 12 months (*n* = 5): median tyrosine pre-study was 490 μmol/L (range 200–600 μmol/L), and for the 12 months continuation study, 430 μmol/L (range 200–600 μmol/L) (Table 3). Three of five children had an improvement in blood tyrosine, although four remained above the target therapeutic range (200–400 μmol/L).

Median phenylalanine concentrations were all ≥ 40 μmol/L (range 30–100 μmol/L) (suggested reference range > 50 μmol/L) [19].

### 3.6. Routine Biochemistry

Routine biochemistry measured at regular clinic reviews showed no abnormal clinical findings for liver (including clotting screen) or renal function tests, which included renal tubular phosphate reabsorption, urine creatinine, calcium and phosphate, microalbumin, urine protein, retinol-binding protein and electrolytes. In addition to routine liver function tests (LFTs), alpha-fetoprotein was measured and was within the reference range (1–10 Ku/L as measured by Roche E601).

### 3.7. Anthropometry

In part 1 (*n* = 9), height, weight, and BMI z-scores remained unchanged. Median z-scores for height were −0.5 (range −2.9 to 0.8), weight, −0.3 (range −1.7 to 1.1) and BMI, 0 (range −2.0 to 2.5). Anthropometry is described in Figure 2 for *n* = 5 children who continued on the study product. After 12 months, height z-scores were improved (median z score −0.3) but was not statistically significant and fell within the population norm.

### 3.8. Gastrointestinal Tolerance

Pre-study: *n* = 3 had abdominal discomfort, *n* = 3 heartburn, and *n* = 2 nausea. One child had a long history of constipation and abdominal pain and was given laxatives, and a further child had a longstanding problem with abdominal pain, which had been extensively investigated. Throughout study parts 1 and 2, there were no significant gastrointestinal problems with the resolution of existing gastrointestinal symptoms for these two children. All other children remained as baseline, with no gastrointestinal disturbances reported.

### 3.9. Palatability

In part 1, 6 of 9 (67%) children who completed at least 24 of the 28-day assessment rated the taste of the study product as favorable, *n* = 3 (33%) were unsure (they neither liked/disliked the study product) but preferred to take their original L-AA supplement post-trial. Seven of nine (78%) children rated the smell favorable.

### 3.10. Ease of Preparation

At the end of study part 1, one teenager (17 y) chose to continue taking a ready-to-drink L-AA supplement as it required no preparation. For the remaining children who preferred the study product, preparation was not a barrier to adherence.

## 4. Discussion

This is the first report to describe the use of a protein substitute based on CGMP in a group of mostly teenagers with HTI. Prior to study commencement, they struggled with protein substitute adherence. Despite the practical difficulties of preparing the study product, 67% (*n* = 6) accepted the new product in study part 1, and 55% (*n* = 5) continued CGMP for 12 months, with 60% (*n* = 3) improving their adherence as measured by tyrosine and phenylalanine concentrations.

The gut function was noticeably improved in two children, one having a long history of abdominal discomfort and the other chronic constipation. These symptoms abated within seven days of starting the product, and both children took all protein substitutes as CGMP. The role of CGMP in modulating gut microbiota and having a prebiotic activity is favorable, although this has not been studied in PKU or HTI subjects on CGMP [9,20].

The pharmacological actions of protein substitutes are often underestimated, but they are of critical importance in influencing clinical outcomes. They provide a source of essential nitrogen necessary to meet physiological and cellular metabolic functions: improving immunology [21] and allowing normal growth [22]. Biochemically, they provide large neutral amino acids (LNAA), which are competitively transported across the blood brain barrier altering brain neurochemistry [23,24]. In a HTI animal model, mice treated with NTBC ± diet had increased plasma and brain tyrosine and lower phenylalanine concentrations, with serotonin significantly negatively correlated with tyrosine concentrations [25]. Aromatic amino acids in the brain function as precursors for monoamine neurotransmitters; tryptophan producing serotonin and tyrosine producing dopamine, norepinephrine, epinephrine. Neurocognitive impairments have been described with increasing frequency in HTI [26,27,28,29], and in this study group, three required extra learning support through educational health care plans (EHCP). Although the exact pathophysiology of this neurodegeneration is unknown, some authors suggest the ratio and balance of LNAA are responsible, while others suggest elevated tyrosine concentrations lead to oxidative damage of the cerebral cortex [27]. Whilst a tyrosine and phenylalanine restricted diet supplemented with protein substitute lowers the brain tyrosine, it is not normalized. In HTI, both Thimm and Naio [27,30] have suggested possible serotonin deficiency as a cause of decreased neurocognition. Naoi demonstrated that high brain tyrosine concentrations inhibited both tyrosine and tryptophan hydroxylase, the rate-limiting enzyme in serotonin deficiency. Thimm measured cerebral spinal fluid, suggesting serotonin deficiency. In rats, high brain tyrosine concentrations have been considered neurotoxic, resulting in oxidative stress, DNA damage and altered mitochondrial energy changes.

The action of a protein substitute based on CGMP may bring additional benefits as it is a rich source of both LNAA (particularly threonine and isoleucine) and sialic acid. This nine-carbon sugar is a structural and functional component of brain gangliosides and has been shown to correlate with the amounts of docosahexaenoic acid (DHA) and total long-chain polyunsaturated fatty acids in the ceramide tail of brain gangliosides [8]. However, the nutritional and biological roles of sialic acid are not fully understood, but in the developing brain, they appear to make an important contribution to the neurological and intellectual outcome. Their potential benefits on amino acid disorders, in which there are disturbances in amino acid transport affecting brain function, warrant further investigation.

At the start and end of the CGMP 12-month study, median height z-scores showed a non-significant improvement, remaining <0 but within the population norm. The evidence on long-term growth in children with HT1 has rarely been reported. A nationwide study from Finland reported low median adult heigh z-scores in both transplanted and NTBC-treated subjects [31]. In contrast, a retrospective study from Birmingham showed an improvement in height z-scores from 8 years of age, reaching population standards by 16 years of age [32].

Lower concentrations of phenylalanine with phenylalanine deficiency are a persistent finding in HTI [5,33]. CGMP contains residual amounts of both phenylalanine and tyrosine, and this additional phenylalanine may be beneficial in helping sustain phenylalanine concentrations. After 12 months of using the study product, 3 of the 5 (60%) children had a median phenylalanine concentration of ≥40 μmol/L and two (subjects 9 and 11) had median phenylalanine concentrations that met recommended clinical target ranges of ≥30 μmol/L. There is evidence to suggest phenylalanine concentrations should be ≥50 μmol/L if measured as a fasting sample in the morning to prevent daytime phenylalanine deficiency [19]. Interestingly, neither the residual phenylalanine (108 mg per day) or tyrosine (33 mg per day) influenced the blood tyrosine or phenylalanine concentrations. This contrasted with the residual phenylalanine in CGMP and its effect on blood phenylalanine control in children with PKU [17].

There are limitations to this study. HTI is a rare disorder with no single center having a large population of subjects. Therefore, we studied only a small number of children with HTI, and although this group of children avoided high protein food choices, natural protein intake was only approximated. Neither tyrosine or phenylalanine concentrations were systematically monitored in study part 2. Fifty-six percent (*n* = 5/9) of children continued the study product, with the remainder continuing their original protein substitute, demonstrating the need for variety to meet the needs of a small but clinically vulnerable population.

## 5. Conclusions

The CGMP protein substitute designed for use in HTI was well accepted, with no deterioration in metabolic control or growth in a small group of patients. The potential long-term benefits of using a bioactive protein substitute on gut microbiota and phenylalanine concentrations in this group of subjects need to be systematically investigated, as it appears that both gut function and phenylalanine concentrations could be improved.

## Figures and Tables

**Figure 1 nutrients-13-03224-f001:**
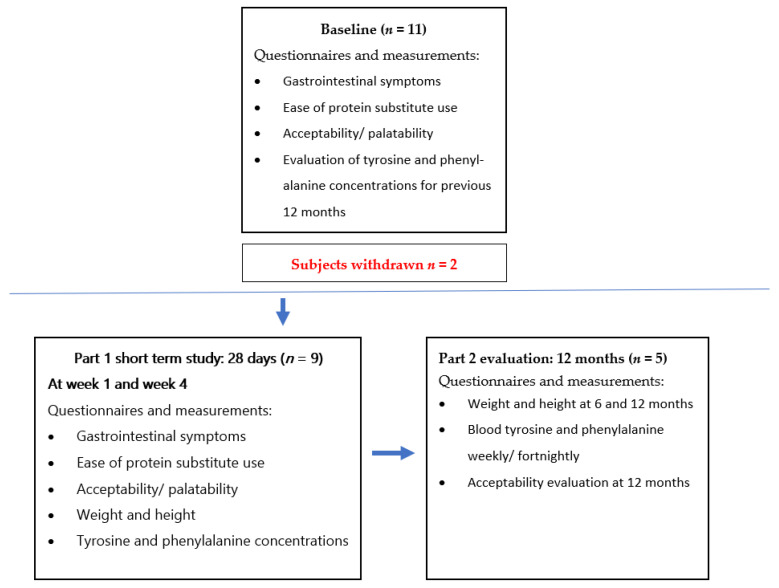
Diagram showing study design. Part 1: a short-term acceptability and tolerance study; Part 2: 12-month follow-up study in HTI children using a study product based on CGMP.

**Figure 2 nutrients-13-03224-f002:**
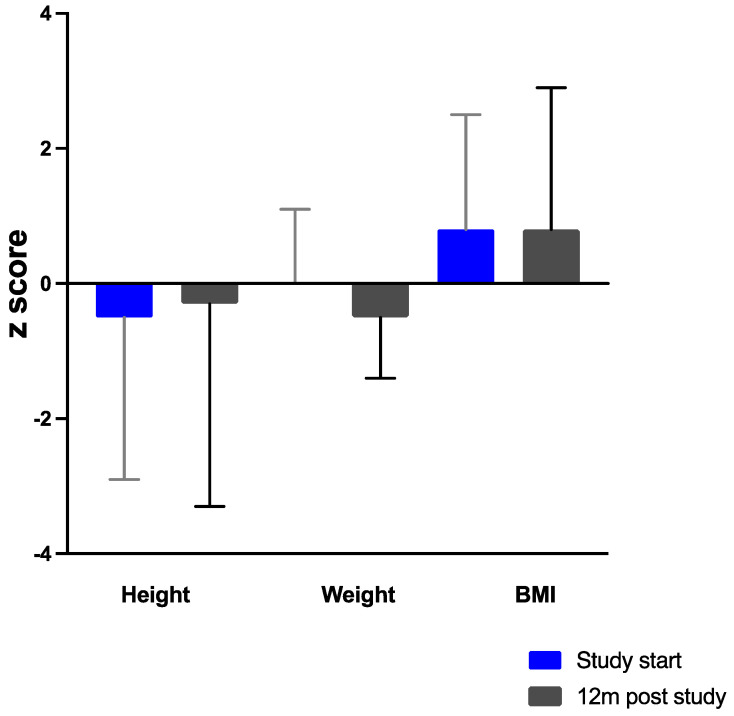
Median height, weight and BMI z-scores for *n* = 5 children at the start and end of the 12 months of using the study product.

**Table 1 nutrients-13-03224-t001:** Nutritional composition of protein substitutes for HTI used in the study.

		Low Phe/TyrCGMP	Phenylalanine/Tyrosine-Free Amino Acid Supplements (L-AA)		
	Units	CGMP PS per 20 g PE	Tyr Cooler 20 g PE	Tyr Express 20 g PE	Tyr Shake and Go 20 g PE
Manufacturer		Vitaflo International	Vitaflo– International	Vitaflo– International	Galen Ltd
**Macronutrients**					
Energy	kJ	508	549	429	693
	kcal	120	130	101	163
Fat	g	1.6	1.6	0.07	<0.5
of which saturates	g	0.35	0.3	0	<0.5
of which DHA	mg	110	134	NA	NA
Carbohydrate	g	6.3	8.9	4.7	17
of which sugars	g	1.4	5.9	0.33	<12
Protein equivalent	g	20	20	20	20
Fiber	g	0	0	0	1.7
Salt	g	0.71	0.26	0.44	0.4
**Vitamins, minerals and micronutrients**					
Vitamin A	µg RE	259	261	283	241
Vitamin D	µg	5.0	10	4.5	4.7
Vitamin E	mg αTE	5.3	5.2	5.3	5.0
Vitamin C	mg	26	37	36.7	30.5
Vitamin K	µg	23	24	34	33
Thiamin	mg	0.6	0.7	0.7	0.7
Riboflavin	mg	0.6	0.8	0.8	0.8
Niacin	mg	3.2	3.5	8.4	8.6
Vitamin B6	mg	0.6	0.9	1.0	0.8
Folic Acid	µg	102	101	136	103
Vitamin B12	µg	1.6	1.6	1.6	1.3
Biotin	µg	13	13	64	52
Pantothenic acid	mg	2.0	1.9	2.7	2.4
Choline	mg	200	200	204	198
Sodium	mmol	12	4.5	7.5	7.9
Potassium	mmol	5.9	6.1	8.0	8.9
Chloride	mmol	0.2	3.9	6.9	7.9
Calcium	mg	399	400	407	371
Phosphorus	mg	413	357	363	293
Magnesium	mg	115	110	128	103
Iron	mg	7.4	7.3	7.3	6.9
Copper	µg	0.6	0.7	0.8	0.6
Zinc	mg	7.4	5.6	7.3	5.7
Manganese	mg	0.4	0.5	1.1	1.1
Iodine	µg	84	85	86	81
Molybdenum	µg	20	23	49	60
Selenium	µg	30	26	30	28
Chromium	µg	12	14	30	24
**Amino acids g/20 g protein equivalent**					
L-Alanine	g	0.83	1.62	1.44	1.32
L-Arginine	g	1.70	1.98	1.85	1.58
L-Aspartic Acid	g	1.31	3.06	2.86	2.75
L-Cystine	g	0.24	0.73	0.69	0.74
L-Glutamine	g	2.70	0.00	1.83	1.8
Glycine	g	2.48	1.62	1.50	2.46
L-Histidine	g	0.70	1.08	1.01	0.86
L-Isoleucine	g	1.42	1.79	1.68	1.48
L-Leucine	g	3.00	2.89	2.69	2.57
L-Lysine	g	0.95	2.05	1.91	1.58
L-Methionine	g	0.28	0.47	0.43	0.40
**L-Phenylalanine**	g	**0.036** **(36 mg)**	**0.00**	**0.00**	**0.00**
L-Proline	g	1.60	1.65	1.55	1.72
L-Serine	g	1.01	1.27	1.18	1.00
L-Threonine	g	2.29	1.39	1.29	1.72
L-Tryptophan	g	0.40	0.57	0.54	0.46
**L-Tyrosine**	g	**<0.011** **(11 mg)**	**0.00**	**0.00**	**0.00**
L-Valine	g	1.14	1.97	1.83	1.75

Legend: DHA docosahexaenoic acid PE, protein equivalent; low Tyr/Phe CGMP, low tyrosine and phenylalanine casein glycomacropeptide (study product); L-AA phenylalanine/tyrosine-free amino acid supplements, Tyr cooler, Tyr express (Vitaflo International); Tyr Shake and Go (Gaelan Ltd. Craigavon, Northern Island, UK), mg milligram of phenylalanine and tyrosine per 20 g of protein equivalent.

**Table 2 nutrients-13-03224-t002:** Subject characteristics and numbers recruited to study parts 1 and 2.

PART 1 SUBJECTS SHORT TERM EVALUATION OVER 28 DAYS							
**Subject**	Age (years)	Sex	Number of Estimated Protein Exchanges	Ethnicity	% (Number) of Days Completed in Part 1	Subjects Completing Part 2	% of Protein Equivalent from CGMP
1	17.1	M	28	British Asian	89% (25)	No	100%
2	10.5	M	25	British Asian	86% (24)	No	33%
3	17.7	F	26	Caucasian	100% (28)	No	33%
4	14.7	M	16	Caucasian	64% (18)	No	33%
5	5.3	F	12	Caucasian	1 Withdrawn	No	44%
6	15.5	M	25	British Asian	17 Withdrawn	No	33%
**PART 2 SUBJECTS FOLLOWED UP FOR 12 MONTHS**							
7	15.3	F	24	Caucasian	100% (28)	Yes	100%
8	15.4	M	30	Arabic	100% (28)	Yes	100%
9	13.9	M	20	British Asian	100% (28)	Yes	100%
10	15.0	F	20	British Asian	100% (28)	Yes	100%
11	8.6	F	20	British Asian	100% (28)	Yes	100%

Legend: Subjects 5 and 6 withdrew from the study, F female, M male. CGMP PS, low tyrosine and phenylalanine casein glycomacropeptide (study product).

**Table 3 nutrients-13-03224-t003:** Median tyrosine and phenylalanine blood concentrations (range) for study part 1, 28 days and part 2, follow-up over 12 months when taking the study protein substitute ± amino acid supplements.

Subjects *n* = 9	Median Tyr (Range) μmol/L 12 Months Pre-Study	Median Tyr (Range) μmol/L During 28 Day Study	Median Tyr (Range) μmol/L for 12 Months Using Study Product	Median Phe (Range) μmol/L 12 Months Pre-Study	Median Phe (Range) μmol/L during 28 Day Study	Median Phe (Range) μmol/L for 12 Months Using Study Product
1	312 (200–400)	330 (200–400)	-	45 (30–100)	40 (30–100)	-
2	338 (200–400)	375 (200–400)	-	30 (30–60)	40 (30–60)	-
3	315 (290–400)	324 (290–400)	-	45 (30–60)	55 (50–60)	-
4	428 (200–440)	415 (200–430)	-	40 (30–60)	40 (30–60)	-
7	393 (200–400)	375 (200–400)	430 (290–660)	55 (30–60)	40 (30–60)	40 (30–50)
8	490 (200–500)	460 (200–490)	430 (270–710)	50 (30–100)	50 (30–100)	40 (30–50)
9	535 (200–600)	495 (200–600)	330 (270–380)	55 (30–60)	50 (30–60)	30 (30–40)
10	370 (200–600)	445 (200–600)	570 (320–940)	30 (30–60)	30 (30–60)	50 (30–70)
11	525 (200–600)	385 (200–400)	490 (290–830)	50 (30–70)	60 (30–70)	30 (20–60)
Median	393 (200–600)	385 (200–600)	430 (270 to 940)	45 (30–100)	40 (30–100)	40 (20 to 70)

## Data Availability

The data presented in this study are available on request from the corresponding author.

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
