# Peer review of "Casein Glycomacropeptide: An Alternative Protein Substitute in Tyrosinemia Type I"

_nutrients, 2021, doi:10.3390/nu13093224_

Round 1

Reviewer 1 Report

The study was aimed at evaluating the clinical use of  casein glycomacropeptide as alternative protein substitute in tyrosinemia type I. The subject matter is of interest and paper well written but results are preliminary, therefore it should be considered as short communication.

Specific comments

Abstract: Components present but may need minor writing for readability

Introduction: Hypothesis is clearly presented and it is supported by text

Materials and methods: Procedures are clear, concise, and easily replicable;

Results: Logically presented and answer research hypothesis

Discussion: Statements and conclusions are presented but need minor revision to correlate with data

and link with goals

References: Appropriate to manuscript type and correlated well with text

Author Response

Dear Reviewers please see the file uploaded

Reviewer 2 Report

The Authors have conducted a study on children/teenagers affected by tyrosinemia type 1 with an alternative supplementation with casein glycomacropeptide for short and long period.

MINOR REVISION

The Authors should delete the extra space in the title before the word “TYROSINEMIA”.

Tyrosinaemia type I is mentioned both as HTI (line 10) and HT1 (line 17). The Authors should choose one abbreviation along all the manuscript (lines 14, 29, 30, 36, 54, 62, 64, 75, 78, 125, 127, 142, 181, 259, 268, 275, 281, 287, 303, 312, 319, 320).

The Authors should put a comma after the words “bioactive peptide” at line 12.

The Authors should delete the dot after the word “nitisinone” at line 39 and the comma before the word “prevents” at line 40.

The Authors should rephrase and better specify the concept at line 41.

The Authors should change the font from line 47 to line 65, according to the one of the rest of the manuscript.

The Authors should add a comma after the words “amino acids” at line 53.

The Authors should specify what the abbreviation PKU means at line 57.

At line 63 the study is described as observational, while at line 68 as intervention study. The Authors should clarify the type of study they have conducted.

The text of the paragraph 2.1. Study Design lacks of the reference to Figure 1.

The Authors should improve Figure 1: the squares of Recruitment and Baseline can be joined. It is not clear why Part 1 is in the upper part of the figure, while Baseline and Part 2 in the bottom. It is  not explained the meaning of the line with numbers and arrows in the middle of the figure.

The Authors should delete the repetition at lines 127 and 154 (age and blood, respectively).

The Authors should put MS/MS into brackets after the full name of the technique at line 162 and specify the instrument used.

The Authors should delete the comma after the word “compare” at line 169.

The Authors should check the brackets at line 182.

The Authors should check the legend of Table 2 regarding the subject that withdrew from the study. The Authors should carefully check the paragraph 3.2. Subject withdrawal: it does not fit the the data in Table 2.

The Authors should add a bibliographic reference at the end of the paragraph 3.4. NTBC and concurrent medications.

The Authors should check the number of the Figure at lines 236 and 239.

The Authors should put a comma after the word “children” at line 264 and after the word “tyrosine” at line 286.

The Authors should specify which are the two subjects that “had median phenylalanine concentrations that met recommended clinical target ranges of ≥ 30μmol/L at (lines 305-307). This sentence seems in contrast to what reported in the Results section “Median phenylalanine concentrations were all ≥ 40 μmol/L (range 30 - 100) (suggested reference range >50)” (lines 225 and 226). Please explain.

The Authors should specify why they have studied such a small number of subjects at lines 311 and 312. Are few the children referring to that hospital or is it due to the incidence of the disease in the population, or something else?

In Funding section, the Authors should specify if the sponsor had some role in the design, execution, interpretation or writing of the study.

In the Supplementary tables, the Authors should check for patient ID: in the last 2 tables the initials of the patients are also reported.

Author Response

Thankyou please see the file uploaded

Reviewer 3 Report

Daly et al., have evaluated the effects of casein glycomacropeptide as an alternative to protein substitute in HT1. The study is designed and well executed. However, there are some technical deficits and a few concerns to be addressed. 

  • Fig. 1: Authors mentioned the height in negative score before and after the treatment. Could you please explain and justify?
  • Authors claim that there is no limitation of the study while the small number of subjects itself is a limitation.
  • HT1 may affect several other systems in the body. Did authors check any other parameter in the blood? How about CBC, clotting time and LFT's? These parameters could have revealed a lot of information. If authors have left over samples, they are suggested to analyze more parameters to support their study.
  • Authors didn't discuss the HT1 effects on liver and kidney.  
  • Authors discussed that HT1 affect neurochemistry and cause neurodegeneration. Why didn't you assess the change in neurochemistry or related parameters?   
  • Please revise conclusion and conclude your study concisely.  
  • Please keep the format/font same throughout the manuscript. 

Author Response

Thank you please see the file uploaded

Round 2

Reviewer 3 Report

Authors have addressed some critical concerns and improvised the manuscript.